



# Assimilation of satellite flood likelihood data improves inundation mapping from a simulation library system

Helen Hooker[1], Sarah L. Dance[1,2,3], David C. Mason[3], John Bevington[4], and Kay Shelton[4]

[1]Department of Meteorology, University of Reading, UK
[2]Department of Mathematics and Statistics, University of Reading, UK
[3]National Centre for Earth Observation (NCEO), Reading, UK
[4]Jeremy Benn Associates Limited (JBA Consulting), UK

**Correspondence:** Helen Hooker (corresponding author: helen.hooker@reading.ac.uk), Sarah L. Dance
(s.l.dance@reading.ac.uk), David C. Mason (d.mason@reading.ac.uk), John Bevington (john.bevington@jbaconsulting.com),
Kay Shelton (kay.shelton@jbaconsulting.com).

**Abstract.**

Mitigating against the impacts of catastrophic flooding requires funding for the communities at risk, ahead of an event. Simulation library flood forecasting systems are being deployed for forecast-based financing (FbF) applications. The FbF trigger is usually automated and relies on the accuracy of the flood inundation forecast, which can lead to missed events that

were forecast below the trigger threshold. However, earth observation data from satellite-based synthetic aperture radar (SAR) sensors can reliably detect most large flooding events. A new data assimilation framework is presented to update the flood map selection from a simulation library system using SAR data, taking account of observation uncertainties. The method is tested on flooding in Pakistan, 2022. The Indus River in the Sindh province was not forecast to reach flood levels, which resulted in a non-trigger of the FbF scheme. We found that the flood map selection could be triggered in four out of five sub-catchments

tested, with the exception occurring in a dense urban area due to the simulation library flood map accuracy here. Thus, the analysis flood map has potential to be used to trigger a secondary finance scheme during a flood event and avoid missed financing opportunities for humanitarian action.

     KEYWORDS: Flood inundation, simulation library, forecast-based financing, satellite data, data assimilation.

## 1 Introduction

Our warmer climate is increasing the frequency and intensity of extreme weather events and the exposure and vulnerability of communities and individuals (Pörtner et al., 2022). Large-scale flood forecasting systems predicting flood inundation extent are increasingly used for disaster risk reduction to improve preparedness ahead of a major flooding event (Stephens and Cloke, 2014a; Hooker et al., 2023b; Wu et al., 2020). An ensemble flood forecasting system creates probabilistic flood maps



indicating the likelihood of flooding across a region or country. Flood impact risk factors such as population density, land-use

types or vulnerable infrastructure can also be mapped for the same area. The forecast-flood-likelihood maps can be overlaid with impact maps and depending on the severity of the hazard and the level of impact, a risk profile can be determined. The flood risk profile can be used to inform forecast-based financing (FbF) schemes that enable the pre-release of funds based on the flood forecast, ahead of the flood event (Coughlan de Perez et al., 2015, 2016). Automation of FbF schemes is important for rapid action to take place to mitigate against flooding impacts. The skill of the flood forecasting system is key to *triggering*

the FbF scheme. A *non-trigger* of FbF ahead of or during a flood event might prove catastrophic for those impacted.

Advanced flood forecasting systems, both at global and local levels, link together meteorological and hydrological forecasts of river discharge that drive the selection of pre-computed flood maps from a simulation library (Speight et al., 2021; Hooker et al., 2023a). The use of a simulation library obviates the need to run a hydrodynamic model as part of the forecast process,

reducing computation time and allowing near real-time updating for large areas, which otherwise presents a significant computational challenge. The flood maps within the library are at a relatively higher spatial resolution (e.g. 30 m) compared to the resolution of the driving global hydrological model (e.g. approximately 5 km). This mismatch in scales can lead to problems with flood map selection and can cause gaps where the minimum return period threshold has not been exceeded (a non-trigger) by the forecast discharge (Hooker et al., 2023b). The three main issues that cause this in the global scale model are the represen-

tation of river networks, the return period thresholds determined and the exclusion of dam operations. Rivers that are narrower than a particular width, or catchment areas smaller than a pre-determined size are not resolved by global scale models. In addition, the return period thresholds set may be poorly calibrated due to a lack of ground truth observational data such as river discharge or river water level (Boelee, 2022; Matthews et al., 2022). These two limitations can lead to a non-trigger, i.e. no flood map is selected from the simulation library for a particular sub-catchment. Also, local dam operations such as diversions

of river water for irrigation purposes or rapid releases of flood waters downstream, are not generally included in global scale models. This can lead to over- or under-prediction of forecast discharge, resulting in inaccurate or non-trigger of flood map selection in the forecast.

Satellite-derived observations of flooding have the potential to bring additional spatial information into flood inundation

forecasts compared to in situ point gauging stations. These observations could be used to update and improve the FbF scheme either as part of a secondary finance payment following the acquisition of the satellite data or to improve the flood inundation forecasts going forwards as the flood event evolves. Synthetic aperture radar (SAR) sensors are particularly useful for



remote flood detection, since they can see through cloud, most weather and are active both day and night (Mason et al., 2012; Schumann et al., 2022). Previously, SAR data have been used in several different ways to improve hydraulic models and flood

prediction through data assimilation (DA). DA finds an optimal state (such as water level) and/or model parameter values by accounting for the previous forecast, the observations available, and both of their associated uncertainties. The updated state (analysis) and/or parameter set are used to initiate the next forecast in a feedback loop or cycle. A review of approaches used to assimilate satellite-derived data into hydraulic models (from 2007 until 2015) can be found in Table 7 of Grimaldi et al. (2016) and Table 1 of Revilla-Romero et al. (2016).


When building a new data assimilation system, there are some fundamental choices that have to be made, including the choice of underlying data assimilation method. Several different data assimilation methods have been used for flood inundation studies including ensemble Kalman filters (García-Pintado et al., 2013; García-Pintado et al., 2015; Cooper et al., 2018; Annis et al., 2022; Nguyen et al., 2023), particle filters, (Hostache et al., 2018; Dasgupta et al., 2021b; Di Mauro et al.,

2021, 2022) and variational techniques (Lai et al., 2014; Pujol et al., 2022). We have chosen to use the variational approach in our work. Variational assimilation solves an optimization problem, finding the single solution that maximizes the posterior probability, given the observations and their uncertainties. This is particularly suited to a simulation library system where the set of states ranged over in the optimization process can be limited to the pre-computed solutions in the library.

A second fundamental choice in building a new data assimilation system, is the approach for comparing the model data to the observations. For SAR observations, a first step is often to extract flood extent using an image classification technique (see Section 3, Grimaldi et al. (2016)). The flood extent information could be directly assimilated, as a binary flood map (Lai et al., 2014, e.g.). An alternative is to intersect the edge of the binary flood map with a digital elevation model (DEM) and derive a water level observation (Mason et al., 2007, 2012). SAR-derived WL only provide information at the flood edge and

rely on the spatial resolution and the vertical accuracy of the underlying DEM (Dasgupta et al., 2021a), which makes them difficult to obtain. Nevertheless, water level observations can provide more long-lasting impact than binary flood extents in data assimilation cycling systems (Cooper et al., 2019). Probabilistic flood mapping procedure for SAR data was first introduced by Giustarini et al. (2016). This created the potential for flooding probabilities to be assimilated directly via particle filter approaches (Dasgupta et al., 2021b; Di Mauro et al., 2021, 2022, e.g.). More recently, observation uncertainty associated with

classifying flood extent from SAR data is openly available through the Copernicus Emergency Management Service (CEMS) (Copernicus Programme, 2021). In this article, we develop a new variational approach to assimilate the probabilistic flood



extent data from the CEMS product.

All of these previous studies involved the assimilation of satellite-derived data to update hydrodynamic model states and/or
parameter values by taking a data assimilation cycling approach. The assimilation provides updated initial conditions ahead of
the next forecast cycle. Our new approach differs significantly as we aim to develop a DA framework to assimilate probabilis-
tic flood maps into a simulation library flood inundation forecasting system. We aim to improve the flood map selection by
creating a new analysis flood map. Spatially distributed flood likelihood information derived from SAR data is utilised in the
DA framework. Note that there is no feedback loop to the forecasting system in the DA approach developed. We test the DA
framework using forecast and optical satellite observation data from a major flood event in Pakistan, August 2022 (Floodlist,
2022).

In this article, the flood forecasting system and derivation of the static simulation library along with satellite-derived ob-
servations of flood likelihood are outlined in Section 2. The development of a new DA framework and verification methods
are described in Section 3. Section 4 presents an overview of the 2022 Pakistan flood and details of the data used. The DA
framework successfully triggered flood maps in 4 out of 5 sub-catchments tested as shown in our results, discussed in Section
5. We conclude in Section 6 with recommendations for future work to improve the use of SAR flood likelihood data to update
a simulation library flood forecasting system through data assimilation.

## 2   Simulation library forecasting system and observation data

Flood Foresight, a simulation library flood inundation forecasting system and its application for disaster risk reduction through
FbF, is outlined in Section 2.1. Section 2.2 details the derivation of the flood likelihood data from SAR that will be used as
observation data in the assimilation process. The extraction of flood extent information from optical images that will be used
for validation is explained in Section 2.3.

### 2.1   Flood Foresight and Forecast-based-Financing

Figure 1 shows the chain of flood forecasting systems used to predict flood inundation and to produce flood impact forecasts.
The Global Flood Awareness System (GloFAS) couples global ensemble weather forecasts with a hydrological model and
provides daily ensemble forecast river discharge at approximately 10 km grid size (v3.2, GloFAS (2021)). The GloFAS forecast
river discharge is used to as an input to drive the Flood Foresight system. Flood Foresight (Revilla-Romero et al., 2017; Hooker





et al., 2023a) is a fluvial, probabilistic flood inundation forecasting system. Flood Foresight is set up by dividing the catchment

into 'Impact Zones' (IZ) or sub-catchments using the HydroBASINS data set (Lehner, 2014). Each IZ in Flood Foresight is linked to a GloFAS grid cell that provides a 51 ensemble member forecast of river discharge. Flood Foresight contains a simulation library of precomputed flood depth and extent maps. The flood map library was hydrodynamically modelled using JFlow®, (Bradbrook, 2006) and RFlow using a detailed 30 m digital surface model. The maps were modelled at specific return period (RP) thresholds (20, 50, 100, 200, 500 and 1500 years). Subsequently, these were linearly interpolated at 5 intermediate

intervals between each RP threshold and extrapolated between zero and the 20 year RP flood map (totalling 36 flood maps). Depending on the forecast discharge from GloFAS for each IZ, a flood map is selected from the simulation library. The flood map selected is determined by the RP threshold exceeded within each IZ. The resultant forecast flood map is created by stitching together individual IZ flood maps (at various RPs) and is produced daily out to 10 days ahead.

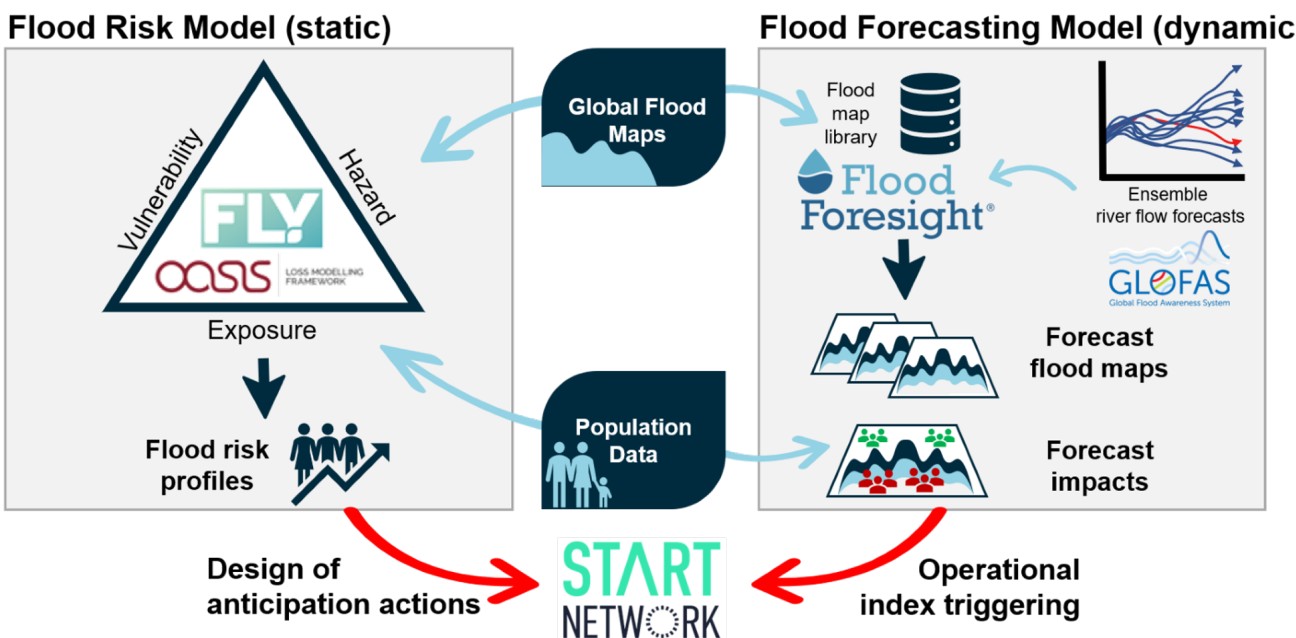

**Figure 1.** Flood Foresight/Start Network ensemble flood inundation forecast and population impacts work flow.

The charity Start Network (Start Network, 2022) brings together a group of over 80 humanitarian agencies and aims to

develop local community-led, early action through a model of proactive funding to mitigate against the impacts of crises. JBA Consulting, in partnership with Start Network, have developed a Disaster Risk Financing (DRF) system for the Indus River basin in Pakistan that links Flood Foresight forecast flood maps to populations impacted by flooding (Fig. 1). For the purposes of setting FbF trigger threshold levels, the DRF system quantifies the flood risk to the population through a probabilistic global





catastrophe risk model, FLY (Dunning, 2019). The dynamic operational index triggering could be linked to the analysis flood

map (produced here as a result of the DA) to provide a secondary finance payment.

## 2.2   Satellite-derived flood likelihood

The GFM service (GFM, 2021) combines the outputs of three different algorithms to extract flood extent and uncertainty information from Sentinel-1 SAR data. The process is automatic and runs continuously in near real-time (within 8 hours after image acquisition) for every SAR image detecting flooding on a global-scale. The resulting flood information layers are available

through open access. The mini-ensemble approach used for flood detection increases the confidence in the resulting derived observations. The first flood mapping algorithm developed by the Luxembourg Institute of Science and Technology (LIST) use a pair of SAR images (pre-flood and flood) and a hierarchical split-based change detection approach to classify permanent and flood waters (Chini et al., 2017). The classification uncertainty depends on the Bayesian inference classification probabilities. The second flood mapping algorithm by the German Aerospace Research Centre (DLR) uses a hierarchical tile-based thresh-

olding approach and the optimization of the classification by combining various information sources using fuzzy-logic theory and region growing (Martinis et al., 2015; Twele et al., 2016). The uncertainty information depends on fuzzy memberships. The final algorithm from TU Wien uses the historical time series of the SAR backscatter values per pixel and classifies flood extent from the backscatter probability distribution (Wagner et al., 2020; Bauer-Marschallinger et al., 2022). The classification uncertainty is based on the Bayesian posterior probability. The output flood layer is derived using the mini-ensemble with a

pixel classified as flooded where two out of three of the algorithms determines a flood class. The flood likelihood values are aggregated, first by converting each to lie in the same range [0, 100] before averaging the likelihood values. Regions where SAR is unable to detect flooding due to shadow or layover effects are removed from the classification process. This usually includes dense urban areas, densely vegetated areas, regions with steep slopes and regions that might appear flooded such as dry, sandy desert-like surfaces. The exclusion mask is available to download as an additional layer. Permanent and seasonal

water bodies are classified separately as a reference water mask layer.

Krullikowski et al. (2023) applied and assessed the usefulness of GFM ensemble likelihood on two test sites in Myanmar and Somalia, both situated in challenging areas for flood detection using SAR data. Krullikowski et al. found that the GFM ensemble likelihood layer resulted in improved trust in the ensemble flood extent detection approach and provides more reliable

and robust uncertainty information for detecting flooding compared to using a single algorithm only.





### 2.3 Optical Normalized Difference Water Index (NDWI)

Occasionally, optical satellite data can be useful for observations of flood extent. Flood detection from optical satellites depends on a near cloud free sky where the satellite acquisition coincides with the flood event. The Normalized Difference Water Index (NDWI) for flood and surface water detection is calculated with Sentinel-2 optical data using the green band (B03) and NIR band (B08) (Albertini et al., 2022). The NDWI is given by:

$$NDWI = \frac{B03 - B08}{B03 + B08},\tag{1}$$

where positive values indicate water. Albertini et al. (2022) reviewed the performance of surface water and flood detection metrics using multispectral satellite data. They found that the average overall accuracy from previous flood studies applying NDWI to be 87.85% and for permanent surface water studies scored 94.41%. This included studies using data from different satellite sensors with spatial resolutions ranging from 10 m for Sentinel-2 to 500 m for Terra-Aqua MODIS.

## 3 Methods

### 3.1 Data assimilation framework

The aim of the DA framework is to update a previous forecast of flood inundation extent and depth from Flood Foresight (the background) where a non-trigger has occurred in the forecast system but where flooding was derived from concurrent satellite-based SAR data. Using observation uncertainty information from the GFM flood likelihood layer (Section 2.2), we aim to improve the flood map selection for non-triggered IZ by minimising a cost function per IZ.

The data assimilation framework aims to update the state vector, $\mathbf{x} \in \mathbb{R}^n$, which contains flood depths at each grid cell location. The components of $\mathbf{x}$, $x_i$, are the individual flood depths for a specific grid cell location. The total number of grid cells across an IZ is $n$, the total of observed unmasked grid cells is $m$. To find the optimum state accounting for observation uncertainty we define the observation *likelihood* term $P(\mathbf{y}|\mathbf{x})$ where the components of observations, $\mathbf{y} \in \mathbb{R}^m$ have two possible binary outcomes, $y_i = 1$ (flooded) and $y_i = 0$ (unflooded), following classification from SAR data. The likelihood term can be represented by a Bernoulli distribution (Lauritzen, 2023), defined as

$$P(\mathbf{y}|\mathbf{x}) = \prod_{i=1}^{m} L_i^{[\mathbf{H(x)}]_i} (1 - L_i)^{1 - [\mathbf{H(x)}]_i},\tag{2}$$





where $L_i$ is the GFM flood likelihood value (see Section 2.2). The $i$-th component of the observation operator $\mathbf{H}(\mathbf{x})$, defined as

$$[\mathbf{H}(\mathbf{x})]_i = \begin{cases} 1 & \text{flooded} & x_i > 0.2m \\ 0 & \text{unflooded} & \text{otherwise} \end{cases}, \tag{3}$$

acts to convert flood depths (state space) to a binary flood class (observation space) at unmasked grid cells. Thus, we exclude observation likelihood information for masked grid cells where SAR data cannot reliably detect flooding. To find the maximum posterior likelihood of the state variable, we take the negative log likelihood of $P(\mathbf{y}|\mathbf{x})$ and divide by the number of unmasked grid cells $m$ to derive the cost function (averaged across unmasked grid cells per IZ)

$$J(\mathbf{x}) = -\frac{1}{m}\sum_{i=1}^{m}\big\{[\mathbf{H}(\mathbf{x})]_i \ln(L_i) + (1 - [\mathbf{H}(\mathbf{x})]_i \ln(1 - (L_i)\big\}. \tag{4}$$

The value of $J(\mathbf{x})$ is calculated per IZ by iterating through the flood map library (36 flood maps) and finding the flood map return period by minimising $J(\mathbf{x})$, which is equivalent to maximising the posterior likelihood, given the observation uncertainty data. To ensure that the minimum is reached across the flood map library, $J(\mathbf{x})$ is calculated for all 36 flood maps. Note that this is different to the standard data assimilation approach where minimisation would be accomplished via a gradient descent algorithm (Bannister, 2017). Following the assimilation process, replacing the non-triggered IZ with updated flood maps results in an analysis flood depth and extent map (the flood depth information is contained within the simulation library). This means that the analysis flood map remains consistent with the Flood Foresight system where the flood maps have been hydrodynamically modelled, i.e. they are physically realistic. Retaining the flood depth information is important for FbF applications for quantifying the risk of flood impacts. Since the observations have binary values (flooded/unflooded), we cannot distinguish between floods that have the same extent but different depths from the observation data. This property is inherited in the cost function.

## 3.2 Validation methods

The resulting analysis flood map, following assimilation of SAR-derived flood likelihood data, is validated by comparing against independent flood extent observation data derived from optical satellite data, the NDWI (Section 2.3). The results will be validated by calculating the Fraction Skill Score (FSS, Roberts and Lean (2008); Hooker et al. (2022)) and by mapping the performance on a Categorical Scale Map (CSM, Dey et al. (2014); Hooker et al. (2022, 2023a)). Both the FSS and the CSM avoid issues with the double penalty impact of conventional binary performance measures as well as the impact of flood


magnitude on the skill score (Hooker et al., 2022). The FSS is based on the Brier Skill Score and uses a neighbourhood approach to determine the skillful spatial scale of the analysis flood map. The fraction of flooding within a given square neighbourhood size of length $n$ is compared by calculating the mean-squared-error (MSE) between the analysis and the validation flood maps to give

$$FSS_n = 1 - \frac{MSE_n}{MSE_{n(ref)}},\tag{5}$$

where $MSE_{n(ref)}$ is a potential maximum MSE that depends on the fraction of flooding across the IZ on the analysis and the validation flood maps. A skilful scale is determined when $FSS \geq FSS_T$, the target FSS score, where $FSS_T \geq 0.5 + \frac{f_o}{2}$ depends on the fraction of observed flooding across the IZ, $f_o$. When the analysis and validation flood extents are equal in area across an IZ there is said to be no background bias and the maximum FSS is 1. Otherwise, the maximum asymptotic FSS (AFSS) is given by

$$AFSS = \frac{2f_o f_a}{f_o^2 + f_a^2},\tag{6}$$

where $f_a$ is the is the fraction of flooding on the analysis flood map per IZ.

The CSM plots a local agreement scale (S) at every grid cell. An overview of the method is presented here. Please see Dey et al. (2014); Hooker et al. (2022, 2023a) for full details of the methodology. A background bias between the analysis and verification flood maps that is deemed acceptable is predetermined. The pre-set bias is used to calculate an agreement

criterion that must be reached by the flood map comparison calculation. The comparison begins at each grid cell $n = 1$, if the agreement criterion is met at grid level, the grid cell is labelled with an agreement scale $S = 0$. Where the criterion is not met, a larger neighbourhood size is compared (e.g. $n = 3$). The fraction flooded in each of the analysis and validation flood maps are compared and if the criterion is met, the agreement scale assigned would be $S = 1$. The process continues to larger

neighbourhoods (e.g. $n = 7, S = 3$) until either the criterion is met or a predetermined limit is reached ($S_{lim}$, set to 9 for this application). The agreement scale at this limit would indicate a false alarm or miss for the grid cell. Note that the relationship between $n$, the neighbourhood size used for the FSS, and $S$ is given by $S = (n-1)/2$. The agreement scales are combined with data from a conventional contingency map (Stephens et al., 2014) and are mapped across an IZ. The CSM indicates a location-specific level of agreement and shows where the flood maps are over- or under-estimating flooding.



## 4    Pakistan flood 2022

### 4.1    Event overview

In Spring 2022, Pakistan experienced a record-breaking heatwave with temperatures exceeding 50°C. The heat exacerbated upstream glacial snow melt feeding the Indus River basin, which runs over 3000 km across the length of Pakistan, draining the Himalayas to the Arabian Sea. An intense monsoon season followed in July and August, driving multiple flood-producing mechanisms including multi-day extreme precipitation that was the primary driver of floods (World Weather Attribution, 2022; Nanditha et al., 2023). Attribution studies indicate that the 5-day maximum rainfall over the provinces Sindh and Balochistan, which led to catastrophic flooding, was made 75% more intense by 1.2°C of global warming (World Weather Attribution, 2022). The northern Sindh province received an estimated 442.5 mm of rainfall in August, 784% more than usually recorded, causing inundation of 55,000 km$^2$ across the region (Floodlist, 2022). Despite early warnings of the potential for significant flooding from GloFAS, the unimaginable scale and magnitude of the flood impacted over 33 million people with over 1700 lives lost and costing more than $40 billion in economic damages (Floodlist, 2022; World Resources Institute, 2023).

### 4.2    Data

The DA framework (Section 3.1) was tested in the northern Sindh province where widespread flooding occurred during August 2022. Figure 2 maps the NDWI derived from Sentinel-2 optical data (Section 2.3) that was used to verify the resultant analysis flood map following DA. Local reports and photographs of flooding were made in the cities of Sukkur and Larkana (DAWN, 2022; The Guardian, 2022; Sky News, 2022). The DA was applied to 5 IZ covering 3 different scenarios (Fig. 3): (1) One IZ where a large proportion of the IZ is a dense urban area and is masked (where the GFM product is currently unable to detect flooding), Sukkur (S), see Figure 3(a); (2) Two IZ with mixed urban and rural areas, Larkana north and south (LN, LS); and (3) two flood edge locations (FE1, FE2). The GFM flood likelihood data used to represent observation uncertainty in the DA is mapped in Figure 3(a) where darker shades of orange indicate a higher likelihood of flooding (Section 2.2). The forecast data from Flood Foresight is mapped in Figure 3(b) where the purple shades indicate the maximum return period flood map triggered by the system from 10 to 31 August, 2022. Each of the IZ selected were non-triggered IZ during this period. The driving forecast river discharge data from GloFAS did not reach the required threshold to trigger a flood map along the central Indus channel. This is likely due to poor calibration of GloFAS due to a lack of observation of river stage or discharge. The Sukkur barrage operations for diversion and altering of river water flows are not currently included in the GloFAS system, which makes forecasts unreliable along this stretch of the Indus River.



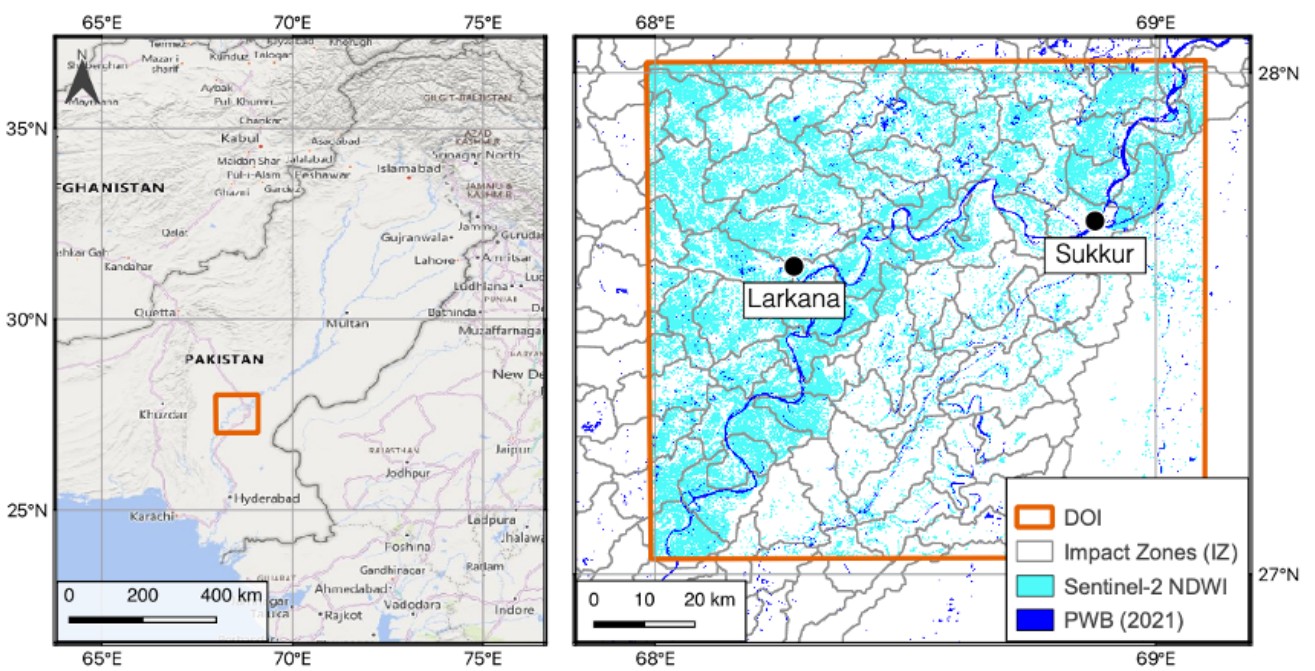

**Figure 2.** The domain of interest (DOI) is located on the Indus Basin, Sindh province, Pakistan (left). Map of the DOI is generated using Bing Maps. ©2024 Microsoft Corporation. The region is divided into sub-catchments or Impact Zones (IZ) in Flood Foresight (right). Satellite-derived flooding (NDWI) from Sentinel-2 data (Section 2.3) from 31 August 2022 is highlighted along with permanent water bodies (PWB).

## 5 Results and discussion

Results are presented for the 3 scenarios tested in the following section. We discuss the benefits and limitations of the approach and how the method could potentially be modified for improved performance.

### 5.1 Scenario 1

The DA framework was applied to 3 different scenarios totalling 5 IZ. The first scenario tested was an IZ centred on the city of Sukkur. Sukkur is located just south of a large barrage, used to control flood waters. Significant flooding was observed locally in the Sukkur region (Sky News, 2022), however the dense city centre means that flooding is difficult to detect using Sentinel-1 imagery at 20 m spatial resolution. Around one third of the IZ is masked by the GFM process (Fig. 4(c)) but high

flood likelihood values are visible across some areas of the IZ (Fig. 3(a)). The aim is to test whether the DA framework can select a flood map from the simulation library based on limited usable SAR data.



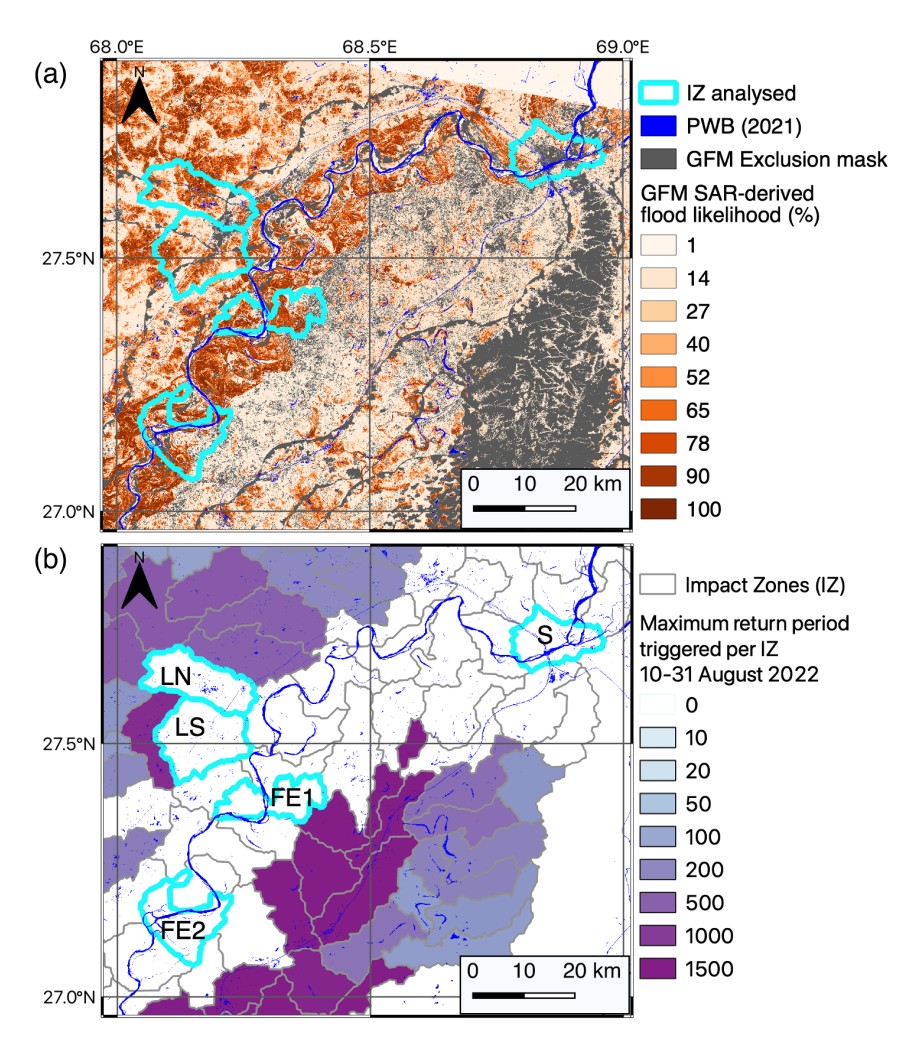

**Figure 3.** (a) GFM flood likelihood derived from Sentinel-1 SAR data, masked areas (grey) indicate where flooding cannot be reliably detected from SAR data. (b) The maximum return period threshold triggered per IZ by the Flood Foresight system during peak flooding 10-31 August, 2022. Five non-triggered IZ of interest labelled: S - Sukkur, LN - Larkana North, LS - Larkana South, FE1 - flood edge 1, FE2 - flood edge 2. Note that PWB means permanent water bodies.

The value of the cost function $J(\mathbf{x})$ from eqn. (4), Section 3.1 is plotted against the RP value of each flood map from the simulation library in Figure 4(a). The cost function was minimised at the lowest RP flood map (3 years) and we found that

a 'no flood' map gave a slightly lower value of $J(\mathbf{x})$. In this instance, the lower RP flood maps over-estimated flooding in areas where low flood likelihood values were derived from SAR. The influence of the Sukkur barrage and river canals running across the IZ made the hydrodynamic modelling more difficult. Also, the flood maps do not include local defence information

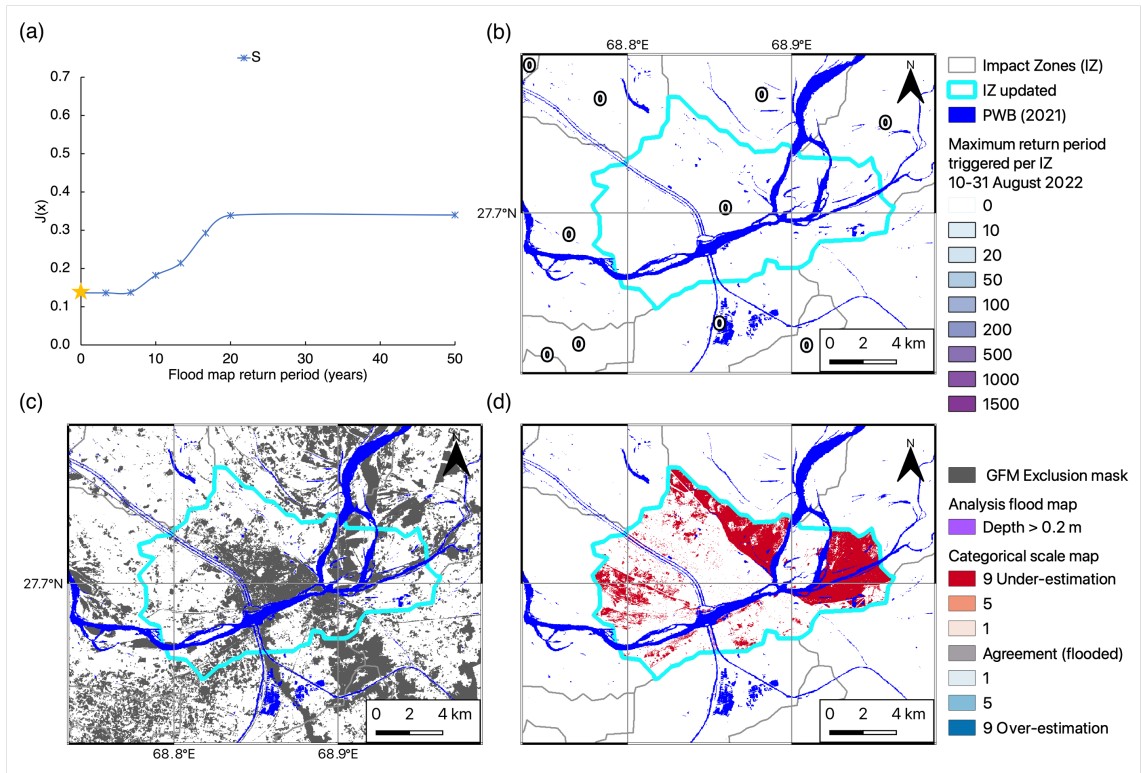

**Figure 4.** Scenario 1: Sukkur, a dense urban area. (a) The cost function (eqn. (4)) plotted against the RP value, the yellow star highlights the minimum value of $J(\mathbf{x})$. (b) The analysis RP triggered following DA. (c) The analysis flood extent map (note that no map was triggered for Sukkur) and (d) the CSM comparing the analysis flood extent map against Sentinel-2 NDWI.

and the flood map interpolation process is highly uncertain at RP less than 20 years. The results mean that no flood map was triggered following the DA (Fig. 4(b and c)) with the CSM map (Fig. 4(d)) indicating where the flooding was underestimated, 265 particularly upstream of the Sukkur Barrage, with no flood map selected.

## 5.2 Scenario 2

The second scenario focused on a mixed urban and rural area with 2 IZ chosen around Larkana city. The dense urban area is again masked and is split across the 2 IZ (Fig. 3(a)), but there are large unmasked areas with high and low flood likelihood values. The assimilation results for scenario 2 are plotted in Figure 5(a) where the cost function minimum value is similar for 270 both LN and LS with a 7 year RP flood map triggered for LN and a 13 year RP flood map triggered for LS (Fig. 5(b)). The resultant analysis flood maps selected (Fig. 5(c)) also indicate flooding within Larkana city, overlapping the masked area. This is consistent with local observations and is important for population impact calculations for FbF schemes. The CSM indicates



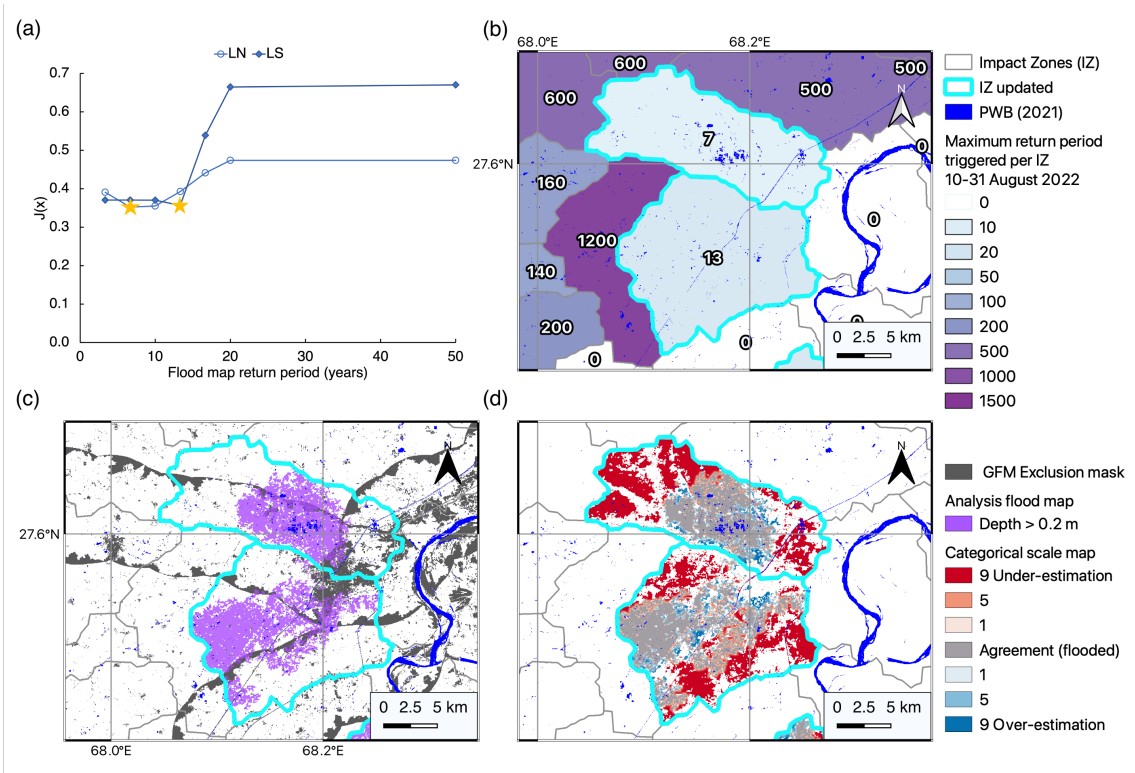

**Figure 5.** Scenario 2: Larkana, a mixed urban and rural area. (a) The cost function (eqn. (4)) plotted against the RP value, the yellow star highlights the minimum value of $J(\mathbf{x})$. (b) The analysis RP triggered following DA. (c) The analysis flood extent map and (d) the CSM comparing the analysis flood extent map against Sentinel-2 NDWI.

that whilst a large area is now correctly indicating flooding there are also large areas that are under-estimated by the analy-

sis flood map (Fig. 5(d)). The neighbouring IZ that were triggered by the forecast system (Fig. 5(b)) are at much higher RP

thresholds than the ones selected following the DA. By inspecting higher RP flood maps than those selected by the DA (for LN

and LS) it became clear that these were over-estimating flooding in locations where low flood likelihood values were located

causing $J(\mathbf{x})$ (Fig. 5(a)) to increase.

One potential solution to the inconsistency seen across the domain could be overcome by including information from the

forecast system. The assimilation process could be carried out across a region including multiple IZ at the same time, rather

than considering individual IZ. Conditions could be imposed, such as a consistent flood depth across IZ boundaries away

from the flood edge. One way to impose some smoothness would be through the use of a background error term. Note that

the background error is the prior or forecast error. Information from neighbouring IZ could be spread across a domain by the



background error covariance (**B**) matrix used in variational DA (Bannister, 2008). The matrix **B** could be calculated offline

using the content of the simulation library.

Once the entire IZ becomes inundated at a 20-year RP, $J(\mathbf{x})$ remains constant with increasing RP (Fig. 5(a)). Although the

depth values are increasing, there are no significant changes in flood extent possible across the IZ, meaning the cost function

cannot distinguish between flood maps over a 20-year RP. For the IZ tested here, the minimum has already occurred at lower RP,

but it is possible that the minimum could occur where $J(\mathbf{x})$ is constant, meaning a range of potential RPs are possible solutions.

In order to distinguish between equally plausible flood maps, additional observation data would be required to measure flood

depth. Flood depth data for sufficiently large floods could be derived from satellite altimetry data such as the Surface Water

and Ocean Topography (SWOT) mission (Frasson et al., 2019; de Moraes Frasson et al., 2023).

### 5.3    Scenario 3

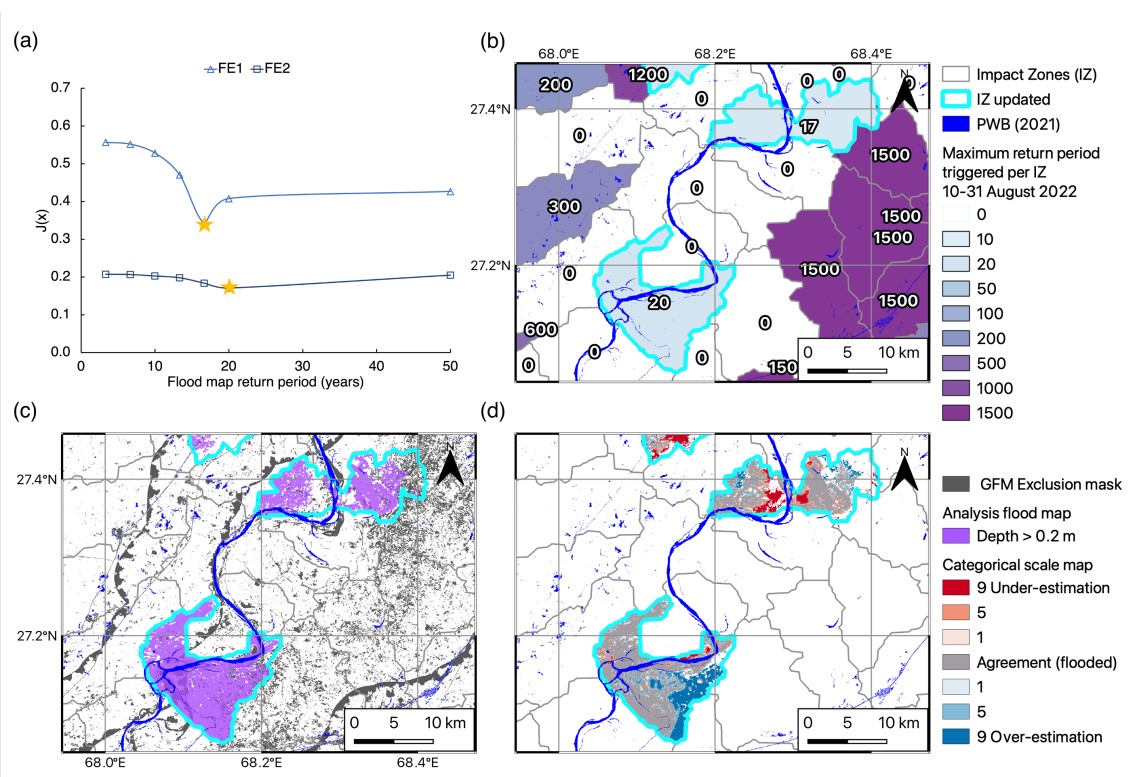

**Figure 6.** Scenario 3: Flood edge location. (a) The cost function (eqn. (4)) plotted against the RP value, the yellow star highlights the minimum value of $J(\mathbf{x})$. (b) The analysis RP triggered following DA. (c) The analysis flood extent map and (d) the CSM comparing the analysis flood extent map against Sentinel-2 NDWI.





The final scenario investigates the impact of assimilating SAR-derived flood likelihood data on flood map selection where
the flood edge lies within the IZ. The cost function value of $J(\mathbf{x})$ drops relatively sharply for FE1 in the north (Fig. 6(a))
from 0.56 at 3 years RP to a minimum of 0.34 at 17 years RP. Further south, $J(\mathbf{x})$ for FE2 is initially lower at 0.21 at 3 years
RP, gradually decreasing to a minimum of 0.17 at 20 years RP. The shape of the cost function shows a smoother descent to a
minimum compared to scenarios 1 and 2 as there is more variation in flood extent between flood maps at different RPs at the
flood edge location. In similarity to scenario 2 (Larkana), neighbouring IZ are again at very high RP levels (Fig. 6(b)). The
analysis flood map for FE1 does not reach the edge of the IZ, whereas FE2 virtually flood fills the IZ (Fig. 6(c)). The CSM
(Fig. 6(d)) shows that some flooding close to the main Indus River has been under-estimated in FE1 but with overall good
accuracy and limited over-estimation. FE2 shows over-estimation in the west but again a large area in agreement with the flood
extent derived from Sentinel-2 NDWI. Using flood extent observation likelihood data would be more useful where the flood
edge stays within the IZ tested for the maximum RP flood map. There would be more chance of variation in the cost function
value across the full range of RP flood maps. In contrast, for a (near) flood filled IZ there would be less variation seen in $J(\mathbf{x})$
due to limited changes in flood extent. Future work could focus on assimilating flood edge IZ and sharing information with
neighbouring IZ by including a background term to update regions closer to the river channel where the IZ are more likely to
be flood filled (Section 5.2).

**5.4    Analysis flood map validation**

**Table 1.** Validation skill scores for each IZ analysis flood map compared against independent Sentinel-2 NDWI

| IZ code | Analysis RP | $FSS$ at $(n=1)$ | $FSS_T$ | $AFSS$ | n at $FSS_T$ |
|---------|-------------|------------------|---------|--------|--------------|
| S | 0 | 0 | n/a | n/a | n/a |
| LN | 7 | 0.32 | 0.64 | 0.40 | $AFSS < FSS_T$ |
| LS | 13 | 0.38 | 0.65 | 0.51 | $AFSS < FSS_T$ |
| FE1 | 17 | 0.55 | 0.66 | 0.71 | n = 35 (525 m) |
| FE2 | 20 | 0.49 | 0.63 | 0.77 | n = 15 (225 m) |

In Table 1 the FSS (Section 3.2) has been calculated by comparing the analysis flood map selected per IZ with the corre-
sponding Sentinel-2 NDWI representing observed flooding, with permanent water bodies excluded from the validation. The
target skill score $FSS_T$ and the asymptotic FSS $AFSS$ are also calculated. The FSS for scenario 1 (Sukkur) was 0 as no flood
map was triggered. For scenario 2 (LN and LS) the FSS score at grid level $n = 1$ (0.32 and 0.38) is around half of $FSS_T$.
Usually, by increasing the neighborhood size the value of FSS also increases, eventually exceeding $FSS_T$. In this case $AFSS$
(which is calculated using the fraction flooded across the IZ from both the analysis and observed flood maps) is less than

$FSS_T$ meaning the total differences in flood extent are too large for the FSS to reach or exceed $FSS_T$. The result of this is that there is not a meaningful or skilful scale of the analysis flood maps for scenario 2. This is due to the under-estimation of flood extent seen on the CSM (Fig. 5(d)). For scenario 3, $FSS_T < AFSS$ which makes it possible to calculate a skilful scale where

$FSS \geq FSS_T$. For FE1 this occurs when $n = 35$ or 525 m and FE2 at $n = 15$ or 225 m confirming that FE2 was the most accurate analysis. These results confirm that future work should focus on flood edge IZ first during the assimilation process.

## 6  Conclusions

In this article we introduced a new DA framework to update and improve the flood map selection within a flood forecasting system designed for FbF applications. Open access flood likelihood data derived from satellite-based SAR is used to update

the flood map selection for previously non-triggered sub-catchments or IZ during a flood event. The framework is tested on the catastrophic flooding in Pakistan, August 2022 for 3 scenarios.

The first scenario tested an IZ where limited useful SAR data was available due to a dense urban area. This resulted in no flood map selection following the DA. The second scenario, where two IZ contained a mix of urban and rural areas did trigger

flood maps but at low RP levels, relative to neighbouring IZ that were previously triggered. This resulted in under-estimation of the flood extent. However, the analysis flood map included flooding across parts of the city of Larkana. Information from the flood likelihood data from other areas of the IZ could select a flood map that included urban flooding. This is useful for FbF applications where population impacts are considered. The final scenario examined flood edge locations and these gave the best results as the variation in flood extent selected higher RP flood maps that were more closely matched to the validation

data. The skilful scale of the analysis flood maps in the flood edge IZ was 225 to 525 m. Out of the 5 IZ tested, 4 resulted in a flood map selection with the dense urban area and limited SAR coverage the exception. Each of these 4 IZ were non-triggered, and now they are, which is beneficial, even if the extents are not perfect.

The non-triggered flood maps could be updated quickly following the production of the GFM flood likelihood layer (approx-

imately 8-hours after SAR acquisition). Although observed flood extent is used in the assimilation, the flood maps selected contain depth values that are already linked to a catastrophe risk model and population impact maps. Therefore the analysis is suitable to inform secondary financing schemes for flood response and recovery, during an event. In this application we were able to analyse the entire flood map library in our variational minimization. For operational applications across a wider area, optimal iteration methods could be used to save computation time and storage. Improvements could also be made by



the inclusion of prior information from the simulation library system. An additional background term in the data assimilation framework could improve the consistency of the flood maps selected across a region.

An additional benefit of our approach is that the analysis flood map could be used in a feedback loop to update the river discharge (e.g. in the associated GloFAS grid cell). This could be useful for hydrological model calibration or in updating the

initial conditions for the next forecast.

The analysis will possibly include surface water flooding (SWF), which is likely for large relatively flat river basins where monsoon rainfall contributes to flooding, such as the Indus basin in Pakistan and the Ganges-Brahmaputra-Meghna catchments of India and Bangladesh. The analysis is more likely to represent flooding as observed 'from the ground', which makes it more

consistent with locally observed flooding. This combined SWF-fluvial analysis flood map would mean that secondary insurance payments are more fairly distributed as they do not depend on the type of flooding mapped within the fluvial simulation library. However, the inclusion of SWF in the analysis flood would cause inconsistencies with the fluvial flood forecasting system used to create the simulation library.

Future options for optimising the simulation library flood maps could use flood depth observations if available, including from opportunistic sources such as camera images (Vandaele et al., 2021, 2023)). In this case, a conventional iterative approach (such as a gradient descent methods) could step through depth values within each individual grid cell. The resultant analysis depth map would represent the best estimate of the *true* flood extent and could be used to inform secondary insurance payments.

*Data availability.* SAR-derived flood layers may be downloaded from the Global Flood Monitoring Portal (GFM, 2022). The forecast flood

maps from the JBA Flood Foresight system are commercial data used under license for this study.

*Author contributions.* JB and KS provided the forecast data. HH wrote the algorithms and ran the experiments, with input from SD, DM, JB and KS. HH prepared the manuscript with contributions from all the co-authors.

*Competing interests.* The authors declare that no competing interests are present.



*Acknowledgements.* This work was funded in part by the Natural Environment Research Council as part of a SCENARIO funded PhD
project with a CASE award from the JBA Trust (NE/S007261/1). SD and DM were funded in part by the EOCIS project. SD also received
funding from NERC National Centre for Earth Observation.





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
