# Peer review of "Assimilation of satellite flood likelihood data improves inundation mapping from a simulation library system"

_Hydrology and Earth System Sciences, 2024_

## Author Comment (AC1)

Many thanks for reviewing our manuscript and for taking the time to provide constructive feedback. We are glad you find our contribution valuable to the disaster mitigation community. We feel it is also a significant contribution to the hydrological community. To the best of our knowledge our manuscript presents the first data assimilation framework to be applied to a simulation library flood inundation forecasting system.

1. Observations are used alongside the model's simulation library flood maps to select the most appropriate model state, or flood map, per sub-catchment. In our opinion, this meets the requirements of a data assimilation approach since it includes both observation and model data. Our cost function has a similar derivation to conventional 3D-Var, but its form is different since, unlike conventional 3D-Var, the observation likelihood is not Gaussian. During the minimization we use a physical constraint, namely that the solution must be a member of the simulation library. This has an analogy to strong-constraint 4D-Var where the solution must fit the forecast model.

2. Thank you for your interest in the evaluation metrics used. The evaluation metrics were developed recently and hold several benefits over conventional binary skill scores and several previous works using them have been cited in the paper (Hooker et al., 2022, 2023a, 2023b). They have been applied here to evaluate the data assimilation performance against independent observation data.

3. Thank you for your useful suggestion. During revision we will add a data assimilation framework flow diagram to aid understanding.

---

## Author Comment (AC2)

*Many thanks for taking the time to read and review our manuscript. Your feedback is appreciated and extremely constructive.*

1) *Thank you, we agree. We propose adding a new diagram to the manuscript illustrating our methodology. Thank you for bringing an additional reference to our attention – we will amend our introduction to discuss and include this paper.*

[Figure]

2) *We agree that the EO data captured the flooding from the Pakistan 2022 event generally very well. For a first demonstration of the method, we wanted a case where there was plenty of observed flooding in contrast to the non-triggered flood maps. However, we also evaluated some sub-catchments where dense urban areas were present such as the Sukkur region and Larkana North and South. The Sukkur sub-catchment has significant proportion masked as unsuitable for SAR-derived flood detection. Whilst the Sukkur sub-catchment didn't trigger a new flood map, this was due to the simulation library accuracy, rather than the size of the masked area. Larkana North and South also contained masked urban areas but in both sub-catchments a flood map was still triggered.*

3) *This is an interesting question and could explain the under-estimation seen in the analysis flood map. A possible approach would be to divide the SAR-derived flood map into two flood maps, fluvial and pluvial, perhaps using a recent detailed DTM. Future work could attempt this and use only the SAR-derived fluvial maps to select from the flood map library.*

*For FbF applications the analysis flood map should include all types of flooding that may cause impacts. A surface water flood map library is available for the UK, Europe and the US but unfortunately, we are not aware of one presently available for Pakistan owing to a lack of a recent detailed DTM. Were this available, a combined system merging fluvial and pluvial flood maps would be ideal to inform insurance payments. On revision we will amend our discussion to reflect this insightful question and our response.*